# Visceral Adiposity Index (VAI) Levels and Metabolic Risk Across Phenotypes of Polycystic Ovary Syndrome (PCOS)

**DOI:** 10.3390/medicina61091673

**Published:** 2025-09-15

**Authors:** Canan Akkus, Oznur Oner, Atilla Okan Kilic, Cevdet Duran

**Affiliations:** 1The Department of Internal Medicine, The Medical School of Usak University, Usak 64300, Türkiye; 2The Department of Obstetrics and Gynecology, The Medical School of Usak University, Usak 64300, Türkiye; oneroznur@hotmail.com; 3The Department of Internal Medicine, The Division of Endocrinology and Metabolism, The Medical School of Usak University, Usak 64300, Türkiye; okankilic@gmail.com (A.O.K.); drcduran@gmail.com (C.D.)

**Keywords:** phenotype, polycystic ovary syndrome, visceral adiposity index, metabolic syndrome, cardiometabolic risk

## Abstract

*Background and Objectives*: Polycystic ovary syndrome (PCOS) is a heterogeneous endocrine disorder often associated with metabolic disturbances such as insulin resistance and metabolic syndrome (MetS). Visceral adiposity index (VAI) is a validated marker that reflects visceral fat distribution and cardiometabolic risk. This study aimed to compare VAI levels among different PCOS phenotypes to evaluate cardiometabolic risk across these phenotypes. *Materials and Methods*: This prospective case–control study included 180 PCOS patients and 51 healthy controls without any metabolic or reproductive issues. Patients were divided into the following subtypes based on the Rotterdam criteria: Phenotype A (*n* = 96), clinical and/or biochemical hyperandrogenism (HA) + oligo-anovulation (OA) + polycystic ovarian morphology (PCOM); Phenotype B (*n* = 19), HA + OA; Phenotype C (*n* = 35), HA + PCOM; and Phenotype D (*n* = 30), OA + PCOM. VAI was calculated for women using anthropometric and biochemical parameters. *Results*: In the total PCOS group, VAI levels were significantly higher than in controls (*p* < 0.001). Phenotypes A and B had higher VAI values than controls and also higher in Phenotype A than in Phenotypes C and D (*p* = 0.003 and *p* = 0.001, respectively). While present in 38 patients (21.10%) in the PCOS group, there was no metabolic syndrome (MetS) in controls (*p* < 0.001). In Phenotypes A, B, and D, while more patients had MetS than controls (*p* < 0.001, 0.004, and 0.021, respectively), more patients had MetS in Phenotype A compared to Phenotypes C and D (*p* = 0.003 and *p* = 0.021, respectively). Given ROC analysis, the VAI cut-off value in predicting MetS in the PCOS group was 1.66 (sensitivity = 94.74% and specificity = 83.10%). *Conclusions*: PCOS phenotypes characterized by HA and OA, particularly Phenotypes A and B, are associated with higher VAI values and an increased frequency of MetS risk. Early identification of these phenotypes may facilitate the implementation of targeted metabolic risk reduction and early intervention strategies, thereby contributing to the reduction of cardiovascular risk.

## 1. Introduction

Polycystic ovary syndrome (PCOS) is one of the most common metabolic disorders and affects 5–20% of women of childbearing age [1,2]. The general characteristics of the condition are oligo-anovulation (OA), clinical or biochemical hyperandrogenemia (HA), and the appearance of polycystic ovaries (PCO) on ultrasonography (USG), and the diagnosis is performed under the Rotterdam criteria [1,2]. Such components of metabolic syndrome (MetS) as obesity, especially abdominal obesity, dyslipidemia, high blood sugar, and hypertension (HT) are frequently witnessed in patients with PCOS [3,4,5,6,7].

Until recent years, only the presence or absence of PCOS was considered significant. However, due to the heterogeneity in clinical features and metabolic disturbances, patients are now classified into four phenotypic groups based on specific criteria: Phenotype A, in which HA, OA, and the appearance of PCO in the ovaries exist on USG (PCOM); Phenotype B, with HA and OA; Phenotype C, characterized by HA and PCOM; and Phenotype D, characterized by OA and PCOM [8]. In PCOS, as the letters of phenotypes progress from A to D, the severity and frequency of accompanying metabolic disorders and inflammation decrease.

Based on the literature, it would be more appropriate to evaluate visceral obesity instead of generalized obesity in predicting cardiovascular morbidity and mortality [9]. Even if PCOS patients are of normal weight, the frequency of central obesity has been reported to be higher in these patients [10]. Compared with generalized obesity, central obesity is more associated with MetS components leading to cardiovascular morbidity and mortality, such as insulin resistance (IR), type 2 diabetes mellitus (DM), HT, and dyslipidemia [11,12,13]. Although waist circumference (WC) provides a simple measure of central obesity, it may be confounded by increases in subcutaneous fat, which has fewer adverse metabolic effects, as well as by variations in ethnicity and gender. Although such methods as computerized tomography (CT) and dual-energy X-ray absorptiometry (DXA) have been recommended to assess the actual amount of visceral fat tissue, the radiation risk and cost of such methods restrict the use of CT and DXA [14,15]. Instead, the visceral adiposity index (VAI), an indicator of visceral fat function calculated from anthropometric [body mass index (BMI) and WC] and biochemical [high-density lipoprotein (HDL) cholesterol and triglyceride (TG)] parameters, has been recommended for early detection and assessment of cardiovascular risk in various populations [16,17,18,19,20]. In one of our previous studies, while finding VAI levels to be similar between non-obese PCOS patients and the controls, we also detected higher VAI values in obese PCOS patients than in obese controls [21].

A recent study demonstrated that VAI values in women with PCOS, particularly those with severe phenotypes, are significantly higher compared to healthy controls [22].

Cross-sectional data have consistently demonstrated that women with PCOS are at an increased risk of cardiometabolic disease compared with those without the condition [23,24]. However, the relationship between specific PCOS phenotypes and cardiometabolic risk determinants, such as VAI, remains unclear and warrants further investigation. Most studies to date have evaluated patients with PCOS without stratifying them into phenotypes. Therefore, this study aimed to address this gap by comparing differences in VAI levels, a key cardiometabolic marker and an indicator of visceral adiposity, both among PCOS phenotypes and between these phenotypes and healthy controls.

## 2. Materials and Methods

The presented study with a prospective and case-controlled design was conducted in the Departments of Internal Medicine, Endocrinology and Metabolism Diseases, and Gynecology and Obstetrics of the Training and Research Hospital at Usak University between 1 June 2022 and 20 July 2023. This study was performed in line with the principles of the 1961 Declaration of Helsinki and its later amendments. Approval was granted by the Ethics Committee of the Medical School of Usak University (date: 25 May 2022, No.: 90-90-13). Informed consent was obtained from all individual participants included in this study. Those admitted to the hospital due to such complaints as pubescence, acne, menstrual irregularity, or infertility and diagnosed with PCOS under the 2003 Rotterdam criteria [1,2] were included in this study, based on the biochemical, hormonal, and USG examinations.

PCOS was diagnosed in the presence of two of these three criteria: (1) OA, (2) clinical (hirsutism and/or acne) and/or biochemical signs of HA, and (3) appearance of PCO on USG, and in the absence of other clinical conditions, such as Cushing’s syndrome, congenital adrenal hyperplasia, or androgen-secreting tumor [2]. The patients were divided into four phenotypes [8]. The subgroup classification criteria for phenotypes A–D were based on the Rotterdam criteria, and all classifications were made solely on the basis of clinical, hormonal, and ultrasonographic findings: Phenotypes A, B, C, and D included 96, 19, 35, and 30 PCOS patients, respectively (Figure 1). Even so, the control group consisted of 51 women aged 18–35 years, age-matched to the PCOS patient group (Figure 1), without diagnoses of PCOS and any known cancers, liver and kidney failures, inflammatory status, taking no medication to affect IR, with no active infection, hirsutism, or acne, with PCO on USG, and having normal menstrual cycles. Height, weight, and WC were measured in all participants, with height and weight recorded in underwear. Hair assessment was repeated by the same researcher (CD). WC was measured as the minimum size between the iliac crest and lateral costal margin. BMI was calculated as weight (kg) divided by height squared (m^2^). To measure the levels of androgen, all blood samples were drawn in the morning after overnight fasting between the 3rd and 5th days of menstruation, separated by centrifugation, and stored in deep freeze at −70 °C until being analyzed. The levels of androgen were measured in none of the controls.

All cases were re-evaluated in terms of PCOM by the same researcher (OO) using a transabdominal USG with a transducer frequency of 3–6.6 mHz and/or a transvaginal USG with a transducer frequency of 3–5 mHz probe using the Mindray DC-7 device (Shenzhen Mindray Bio-Medical Electronics Co., Ltd., Shenzhen, China), and the presence of 12 or more follicles in each ovary measuring 2–9 mm in diameter and/or increased ovarian volume > 10 mL was accepted to be compatible with PCOM.

Glucose levels [normal range (NR), 70–105 mg/dL] were measured by Olympus AU 5800 (Beckman Coulter Inc., Brea, CA, USA) through the hexokinase method. Insulin levels were measured by Immulite 2000 (Siemens Healthcare Diagnostics, Siemens AG, Munich, Germany) with the chemiluminescence method, and the intra- and inter-assay coefficients of the variations for insulin were found as 4.6 and 5.9, respectively. While HDL-cholesterol levels were measured by Olympus AU 5800 (Beckman Coulter Inc., Brea, CA, USA) with the immune reaction (antigen–antibody complex), the levels of TG were assessed with a routine enzymatic method using an auto-analyzer by Olympus AU 5800 (Beckman Coulter Inc., Brea, CA, USA). Even so, the luteinizing hormone (NR: 1.9–12.5 IU/L), follicle-stimulating hormone (NR: 2.59–10.2 IU/L), estradiol (NR: 19.5–144.2 pg/mL), and total testosterone (NR: 12–60 ng/dL) were evaluated through the chemiluminescent immunoassay method on the Abbott Advia Centaur (Siemens Healthineers, Tarrytown, NY, USA) device. In addition, dehydroepiandrosterone-sulfate levels (normal range: 1.9–12.5 IU/L) were measured using the chemiluminescent immunoassay method with the Siemens Immulate 2000 device (Siemens Healthineers, Cary, NC, USA), and 17 OH-progesterone (normal range: <2.6 ng/mL) and free testosterone (normal range: <4.2 pg/mL) were evaluated using the chemiluminescent immunoassay method with the Snibe Maglumi 4000 plus (Snibe, Shenzhen, China) device. However, MetS was diagnosed under the criteria by the National Cholesterol Education Program Adult Treatment Panel (NCEP-ATP) III [25].

While IR was calculated by the Homeostasis Model Assessment—Insulin Resistance (HOMA-IR) [fasting plasma glucose (mmol/L) × fasting serum insulin (µIU/mL)/22.5], VAI was calculated for women by the formula of (WC/[36.58 + (1.89 × BMI)]) × [(TG (mmol/L)/0.81) × (1.52 × HDL-cholesterol (mmol/L))] [17].

## 3. Statistical Analysis

The statistical analyses of data were carried out with the Statistical Package for Social Sciences for Windows, version 22.0 (SPSS, IBM Corp., Armonk, NY, USA). The analyses of data normality were conducted using Shapiro–Wilk tests. In group comparisons, the student *t*-test was used for continuous variables with normal distribution, and the Mann–Whitney U test was used for the variables without normal distribution. While Fisher’s exact test was used for the comparisons of binary groups for categorical variables, Pearson’s chi-square test was used for the comparisons of more than two groups. Additionally, normally distributed continuous variables were given as mean ± standard deviation (SD), and non-normally distributed continuous variables were presented as median (min–max). However, the receiver operating characteristic (ROC) analysis was carried out to calculate the values of sensitivity, specificity, and area under curve (AUC). After the AUC value was found to be significant, the optimal VAI cut-off value for predicting the presence of MetS was determined using the Youden’s J index method. The statistical significance of the findings was evaluated at a 95% confidence interval (CI), and a *p*-value of ≤0.05 was accepted to be significant.

## 4. Results

A total of 231 participants, 180 patients with PCOS, and 51 healthy controls were included in this study. When comparing the total PCOS group, including all PCOS patients, with the controls; BMI levels, weight, WC, insulin, TG, and HOMA-IR levels were found to be higher, and HDL cholesterol levels were detected to be lower (all *p* < 0.001) (Table 1). Additionally, the levels of VAI levels were also higher in the total PCOS group than in the controls [1.21 (0.39–10.89) and 0.85 (0.32–1.87), respectively] (*p* < 0.001) (Table 1, Figure 2).

Compared to PCOS subtypes with controls, BMI levels were determined to be higher in Phenotype A, Phenotype B, and Phenotype D than those in the controls (*p* < 0.001, *p* = 0.028, and *p* = 0.006, respectively), and also higher in Phenotype A compared to Phenotype C (*p* < 0.001) and Phenotype D (*p* = 0.031). However, the levels of WC were higher in Phenotype A (*p* < 0.001) and Phenotype B (*p* = 0.041), compared to the controls, and also higher in Phenotype A than those in Phenotype C (*p* < 0.001) and Phenotype D (*p* = 0.005). Compared to PCOS patients with the controls, Phenotype A was detected to have higher levels of weight, insulin, HOMA-IR, and TG (for all, *p* < 0.001); Phenotype B to have higher insulin and HOMA-IR levels (*p* < 0.001 and *p* < 0.001, respectively); Phenotype C to have higher insulin (*p* = 0.010) and HOMA-IR (*p* = 0.012); and Phenotype D to have higher insulin (*p* = 0.024) and HOMA-IR (*p* = 0.024) levels. Even so, Phenotype A and Phenotype B were found to have lower HDL-cholesterol levels (for both, *p* < 0.001) (Table 1). The levels of insulin, HOMA-IR, and TG were higher in Phenotype A than in Phenotype C (*p* = 0.009, *p* = 0.008, and *p* = 0.010, respectively) and Phenotype D (*p* = 0.006, *p* = 0.008, and *p* = 0.002, respectively). In contrast, HDL-cholesterol levels were lower in Phenotypes C and D (*p* = 0.001 and *p* = 0.021, respectively). In addition, the levels of HDL-cholesterol were lower in Phenotype B than in Phenotype C (*p* = 0.022).

When PCOS subtypes were compared to the controls, VAI levels were observed to be higher in Phenotypes A [1.46 (0.39–10.89)] and B [1.31 (0.43–4.39)] than in the controls [0.85 (0.32–1.87)] (Table 1). When intergroup comparisons were performed among PCOS subtypes, VAI levels were found to be higher in Phenotype A than in Phenotypes C [1.00 (0.41–4.79)] and D [1.01 (0.4–3.21)] (*p* = 0.003 and *p* = 0.001, respectively).

In the presented study, while 38 patients (21.10%) in the PCOS group had MetS, none of those in the controls were found to have MetS (*p* < 0.001) (Table 2). More patients in Phenotypes A, B, and D had MetS compared to the controls (*p* < 0.001, 0.004, and 0.021, respectively). In addition, more patients in Phenotype A had MetS compared to Phenotypes C and D (*p* = 0.003 and *p* = 0.021). Given the findings of ROC analyses, the VAI cut-off value in predicting MetS in the PCOS patient group was found to be 1.66 (sensitivity = 94.74%, specificity = 83.10%, AUC 95% CI = 0.952 (0.937–0.986), standard error (SE) = 0.0153, Youden’s J index = 0.7784, and *p* < 0.001) (Figure 3a,b).

When the correlation analysis was performed, there was a positive correlation between VAI and age levels (Spearman’s Rho *=* 0.214, *p* = 0.001) and HOMA-IR (Spearman’s Rho = 0.348, *p* < 0.001) (Table 3).

## 5. Discussion

In the presented study, VAI levels were evaluated across different PCOS phenotypes. VAI was higher in the total PCOS group, as well as in Phenotypes A and B compared to controls, and higher in Phenotype A compared to Phenotypes C and D. Additionally, the prevalence of MetS was greater in the total PCOS group and in Phenotypes A, B, and C than in controls. The VAI cut-off value for predicting MetS in PCOS patients was calculated as 1.66.

IR and MetS components are frequently present in patients with PCOS [3,4,5,6,26]. Moreover, IR is considered one of the specific features of PCOS, independent of adiposity [7]. On the other hand, hyperinsulinism existing in PCOS leads to a vicious cycle in these patients, triggering hyperandrogenism [26]. Due to the heterogeneity of clinical features and metabolic disturbances, PCOS patients are classified into distinct phenotypes. Metabolic complications are more common in Phenotype A, which represents the classic PCOS features, and in Phenotype B, characterized by HA and OA. The severity and frequency of metabolic disturbances and inflammation progressively decrease from Phenotype A to D [8]. Patients with PCOS tend to accumulate more central and visceral fat, particularly in the trunk and upper body, compared to BMI-matched controls. This pattern contributes to increased proinflammatory activity, IR, DM, dyslipidemia, and HT, ultimately elevating the risk of atherosclerosis and cardiovascular mortality [17,27]. Vitamin D is a secosteroid hormone, and increasing evidence supports its deficiency as a contributor to metabolic disturbances in women with PCOS. Moreover, vitamin D supplementation appears to exert beneficial effects on IR in this population [28]. Myo-inositol (myo-Ins) and its isomers (D-chiro-inositol) have consistently demonstrated beneficial effects on clinical and metabolic features of PCOS. In their study, treatment with inositols has shown improvement of insulin resistance, reaching a reduction of compensatory hyperinsulinemia and improvement of metabolic and ovulatory features in patients with PCOS [29]. Furthermore, alpha-lipoic acid (ALA), an antioxidant with anti-inflammatory, immunomodulatory, and insulin-sensitizing properties, has also been proposed as a potential adjunct therapy due to its favorable impact on insulin resistance. Since ALA improves IR, it has been suggested that ALA could be beneficial in the treatment of PCOS. Although a few studies suggest that combining ALA with inositols may enhance therapeutic efficacy, current evidence remains insufficient [30]. Although WC provides a simple measure of central obesity, VAI has been shown to better reflect cardiometabolic risk by distinguishing between subcutaneous and visceral adiposity [17].

In the presented study, VAI levels were also found to be higher in all PCOS patient groups and Phenotype A and B subtypes than in the controls and also in Phenotype A compared to Phenotypes C and D (Table 1, Figure 2). In the study performed by Amato et al. [31], on a total of 220 Sicilian PCOS patients (82 with Phenotype A, 43 with Phenotype B, 52 with Phenotype C, and 43 with Phenotype D) and 144 controls, VAI levels were found to be 2.45 ± 1.63, 2.49 ± 1.46, 1.68 ± 1.0, 2.25 ± 1.40, and 1.62 ± 0.84 in Phenotypes A, B, C, D, and controls, respectively. In the same study, the researchers also reported that oligomenorrheic phenotypes had higher VAI levels. The levels of VAI determined by Amato et al. [31] are 1.5–2 times higher than those found in our patient and control groups, which may be attributed to the fact that both studies were conducted in different ethnic groups. In a cross-sectional study of 189 young adults in Mauritius, individuals of South Asian descent (Indians) exhibited a more adverse fat distribution pattern and lipid risk profile compared to those of predominantly African descent (Creoles), despite similar BMI and waist circumference [32]. Indices of visceral adiposity and visceral-to-peripheral fat ratios were significantly higher in Indians, and these differences partially explained the less favorable lipid profile, with the visceral-to-peripheral adiposity ratio showing a stronger association with sex- and ethnicity-related cardiovascular risk differences than visceral adiposity alone [32]. In another study by Agrawal et al. [33], in which VAI levels were evaluated in 100 PCOS patients with different phenotypes and 50 healthy controls, it was reported that VAI levels were higher in all PCOS patients than in controls and also higher in Phenotype A than in Phenotype D, which is consistent with our study findings (VAI levels in all PCOS cases: Phenotypes A, B, C, and D, and controls: 2.07, 2.46, 2.48, 1.47, 1.7, and 1.27, respectively). When patients were classified by cardiometabolic risk, Agrawal et al. reported that 56% were in the risk group and 12% in the high-risk group overall. Among Phenotype A cases, 64% were in the risk group and 24% in the high-risk group; for Phenotype B, 17% and 50%; for Phenotype C, 67% were in the risk group; and for Phenotype D, 53% and 2%, respectively [33]. However, the small number of patients in their study, especially in Phenotype B and Phenotype C (six and three patients, respectively), makes it difficult to analyze this study accurately. In addition, the lack of comparisons of VAI levels in PCOS subtypes with the controls can be considered another limitation. In a study of 129 women undergoing in vitro fertilization, Vale-Fernandes et al. [34] investigated the independent effects of PCOS and obesity on reproductive and metabolic parameters. Participants were divided into four groups based on BMI and PCOS status. PCOS was associated, independent of BMI, with higher anti-Müllerian hormone levels, an increased luteinizing hormone/follicle-stimulating hormone ratio, and lower follicular fluid progesterone. Obesity was linked to reduced sex hormone-binding globulin levels and higher HOMA-IR, while women with both PCOS and obesity had markedly elevated androstenedione and testosterone levels. The authors corroborate our findings that PCOS phenotypes characterized by HA and elevated VAI are associated with heightened metabolic and reproductive risks [34]. The independent and synergistic effects they observed highlight the importance of distinguishing the metabolic contributions of PCOS and excess weight when assessing patient risk and individualizing treatment strategies. These results further reinforce the clinical relevance of metabolic markers such as VAI, independent of BMI, in identifying women at increased risk for reproductive and cardiovascular complications, aligning with our emphasis on the role of VAI.

In the study by Ramezani Tehrani et al. [35], although women with PCOS were older and had higher BMI and waist circumference compared to healthy controls, VAI levels were found to be similar between PCOS subgroups and the control group. Considering that the prevalence of obesity and visceral adiposity increases with age [36] and that both BMI and WC levels are the components of the VAI formula, it can be interpreted that such differences affected the study findings.

In our study, the HOMA-IR index, an indicator of IR in the total PCOS patient group and all phenotypic subtypes, was found to be higher than the control group. Upon the comparisons of subtypes, HOMA-IR levels were detected to be higher in Phenotype A than in Phenotypes C and D; even so, no difference was found in HOMA-IR levels among other subtypes. Phenotypes A and B are severe subtypes of PCOS [33,37], and as PCOS subtypes progress from A to D, the severity of metabolic disorders decreases. In their study, although not attributing to HOMA-IR levels in terms of PCOS subtypes, Agrawal et al. emphasized a strong correlation between VAI and HOMA-IR levels as a result of correlation analysis (*r* = 0.455, *p* < 0.001) [33]. On the other hand, Amato et al. reported that HOMA-IR levels were significantly higher only in Phenotype B compared to controls (*p =* 0.037), while Phenotype C demonstrated a lower Matsuda insulin sensitivity index (*p* = 0.02) [31]. As opposed to the findings by Amato et al., Ramezani Tehrani et al. reported that when HOMA-IR levels were taken as ≥2.3, IR was present in more patients in all three PCOS subgroups compared to controls, although HOMA-IR levels in PCOS subtypes were similar to those in the controls.

In our study, while 38 patients (21.10%) in the total PCOS group had MetS, no MetS was detected in the control group (*p* < 0.001) (Table 2). When the subtypes were examined, the frequency of MetS was 31.30% in Phenotype A, including the component of OA; 15.80% in Phenotype B; and 10% in Phenotype C, and these rates were higher than that in the controls (*p* < 0.001, *p* = 0.004, and *p* = 0.021, respectively). In addition, given the intergroup comparisons of PCOS subtypes, more patients in Phenotype A had MetS than in Phenotypes C and D (*p* = 0.003 and *p* = 0.021). When the findings of ROC analysis were investigated, we found the VAI cut-off value as 1.66 (sensitivity = 94.74% and specificity = 83.10%) in predicting MetS in the PCOS patient group (Figure 3a and b). Amato et al. [31] reported that the frequency of MetS was significantly higher only in Phenotypes A (*p =* 0.005) and B (*p* = 0.024) compared to controls. They also observed that MetS prevalence was higher in OA-containing phenotypes, consistent with our study findings. Although MetS prevalence was elevated in Phenotype D as well, this difference did not reach statistical significance. Furthermore, a limitation of their study was the absence of direct comparison of MetS prevalence among PCOS subtypes.

In planning this study, our primary aim was to compare VAI levels among PCOS phenotypes; therefore, the patients in this study were appropriate for our inclusion criteria and the predicted number of patients. After adding the control group, 51 cases age-matched with the PCOS patients and meeting the inclusion criteria were included in this study. In the study by Shreenidhi et al. [38], which compared VAI and lipid accumulation product (LAP) levels in PCOS patients classified as metabolically healthy (MH-PCOS) and metabolically unhealthy (MU-PCOS) according to NCEP-ATP III criteria, 38% of patients were in Phenotype A, 3% in Phenotype B, 12% in Phenotype C, and 47% in Phenotype D. MU-PCOS cases were most frequently observed in Phenotypes A (43%) and D (36.30%). Shreenidhi et al. also found the VAI cut-off value to be ≥2.767 [sensitivity *=* 84.09%, specificity *=* 85.26%, AUC = 95% CI: 0.89 (0.82–0.95)] in distinguishing MU and MH-PCOS in the Asian–Indian population and reported that the risk of having MetS increased by 9.42 (95% CI: 3.25–27.26) times above this value. In their study where VAI levels were compared, Shreenidhi et al. found no difference between the subtypes (from Phenotype A to D: 2.3, 2.1, 1.57, and 2.12, respectively), and no relationship between VAI and HOMA-IR levels [38]. On the other hand, Ramezani Tehrani et al. reported the VAI cut-off value as 3.1 (sensitivity = 81% and specificity = 78%, AUC = 0.8) in predicting MetS [35].

As parallel to the positive correlation between VAI and HOMA-IR levels reported in several previous studies [33,39,40], there was also a positive correlation between VAI and HOMA-IR (Spearman’s Rho = 0.348, *p* < 0.001) levels in our study. However, while finding no relationship between VAI and HOMA-IR (*r* = 0.12, *p* = 0.08) levels, Shreenidhi et al. [38] detected a relationship between LAP and VAI (*r* = 0.23, *p* = 0.001) levels.

When the correlation analysis was performed, there was also a positive correlation between VAI and age levels in our study. The positive association observed between age and VAI in our study is consistent with previous research, as several studies have demonstrated that visceral adiposity increases markedly with age, independent of total body fat, and is strongly associated with heightened cardiometabolic risk [41]. Furthermore, in a nationally representative cohort of US adults, Qiushi Sun et al. [42] demonstrated a nonlinear association between VAI and long-term all-cause mortality, with age-related disparities. Higher VAI levels were associated with an increased mortality risk in younger adults, an effect that diminished with advancing age, underscoring the importance of maintaining optimal VAI for long-term health outcomes, particularly in younger populations.

Potential limitations of our study include the relatively small sample sizes in certain PCOS subgroups (particularly Phenotype B), the limited number of participants in the control group, and the absence of MetS cases among controls, which may restrict the generalizability of the findings. Furthermore, the strict exclusion criteria ensured the homogeneity of the PCOS and control groups; however, the lack of BMI matching in addition to age matching represents a limitation of our study. The other limitation of our study is the absence of assessment of other key cardiometabolic markers in addition to VAI. The cross-sectional design of our study also limits the assessment of causality.

Future studies with large sample sizes comparing VAI with other cardiometabolic markers in PCOS patients should aim to elucidate their respective roles in identifying and managing cardiovascular disease risk among women of reproductive age.

## 6. Conclusions

This study underscores the clinical relevance of VAI in the metabolic assessment of PCOS subtypes, particularly in Phenotype A. We evaluated VAI levels across different PCOS phenotypes in an effort to elucidate the cardiometabolic differences among these phenotypes and found that subtypes containing components such as HA and OA exhibited higher VAI values than both controls and other phenotypes. These groups also showed a greater prevalence of adverse metabolic conditions, including MetS and IR. Our findings highlight the importance of early identification and metabolic risk stratification in PCOS phenotypes to prevent long-term cardiometabolic complications in PCOS, especially in phenotypes characterized by HA and OA.

## Figures and Tables

**Figure 1 medicina-61-01673-f001:**
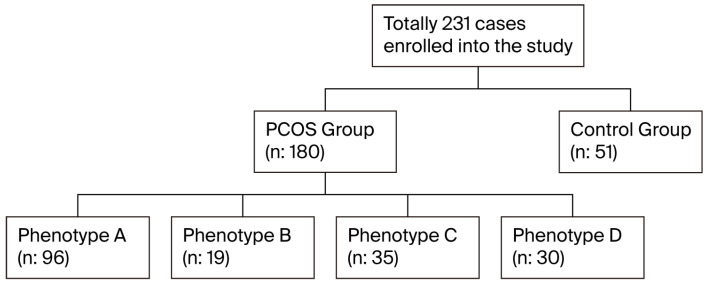
Flowchart of the study population.

**Figure 2 medicina-61-01673-f002:**
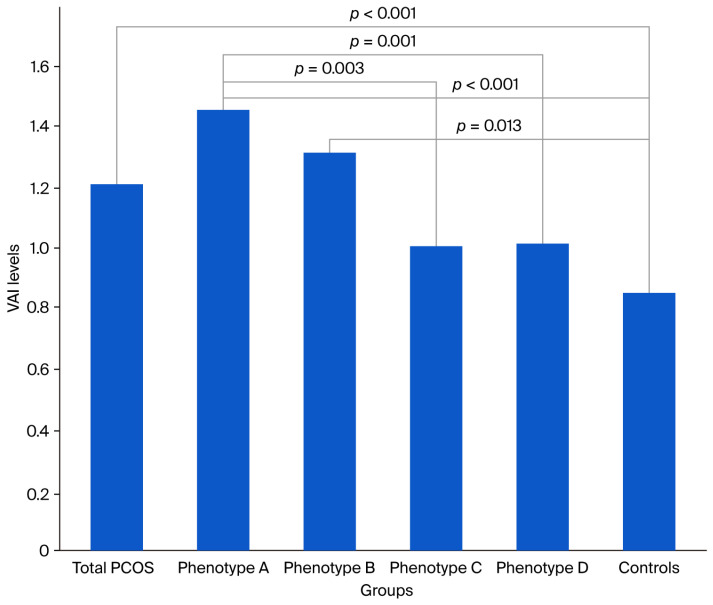
Visceral adiposity index (VAI) levels in total polycystic ovary syndrome (PCOS) patients and PCOS subtypes.

**Figure 3 medicina-61-01673-f003:**
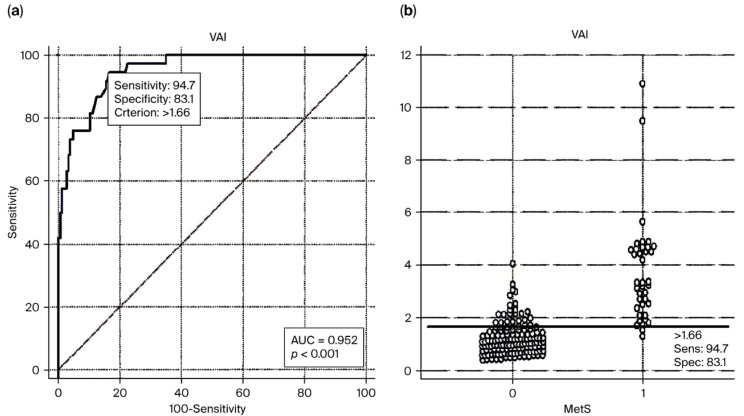
(**a**) The receiver operating characteristic (ROC) curves of visceral adiposity indexes (VAI) in predicting metabolic syndrome (MetS) in patients with polycystic ovary syndrome (PCOS). (**b**) Scattering graph of VAI, according to the determined VAI cut-off value.

**Table 1 medicina-61-01673-t001:** Some demographic and laboratory data of the cases.

	Total PCOS *n* = 180	Phenotype A *n* = 96	Phenotype B *n* = 19	Phenotype C *n* = 35	Phenotype D *n* = 30	Control Group *n* = 51	*p*1	*p*2	*p*3	*p*4	*p*5	*p*6	*p*7	*p*8	*p*9	*p*10	*p*11
Age (year)	24 (18–35)	25 (18–35)	22 (18–34)	25 (18–35)	24 (20–35)	23 (22–33)	0.457	0.191	0.191	0.950	0.408	0.103	0.353	0.954	0.322	0.099	0.500
Weight (kg)		70.5 (45–129)	65 (44–103)	60 (44–98)	63.5 (44–116)	58 (47–79)	<0.001	<0.001	0.065	0.351	0.025	0.082	<0.001	0.022	0.273	0.805	0.304
BMI (kg/m^2^)	25.49 (15.59–50.22)	27.19 (17.15–49.77)	24.83 (15.78–39.84)	22.14 (15.59–34.08)	24.88 (17.19–50.22)	21.48 (17.78–27.34)	<0.001	<0.001	0.028	0.146	0.006	0.111	<0.001	0.031	0.174	0.766	0.184
WC (cm)	83.5 (58–129)	89 (61–129)	82 (60–120)	77 (58–105)	78 (60–116)	75 (59–100)	<0.001	<0.001	0.041	0.237	0.160	0.065	<0.001	0.005	0.269	0.689	0.562
Glucose (mg/dL)	89 (56–142)	90 (56–142)	88 (75–109)	88 (66–103)	88 (78–103)	87 (73–98)	0.082	0.054	0.204	0.588	0.327	0.976	0.280	0.521	0.462	0.622	0.732
Insulin (μU/mL)	12.3 (2–100.5)	13.6 (3.6–100.5)	13.5 (4.4–35.7)	10.1 (2.5–37.8)	9.6 (2–30.5)	7.3 (2.8–22)	<0.001	<0.001	<0.001	0.010	0.024	0.810	0.009	0.006	0.094	0.071	0.797
HOMA-IR	2.75 (0.38–25.57)	2.96 (0.77–25.57)	3.06 (0.92–7.31)	2.30 (0.55–7.54)	2.13 (0.38–6.55)	1.63 (0.58–4.35)	<0.001	<0.001	<0.001	0.012	0.024	0.775	0.008	0.008	0.094	0.071	0.813
HDL-cholesterol (mmol/L)	1.4 (0.9–2.4)	1.3 (0.9–2.4)	1.4 (0.9–1.9)	1.6 (1–2.2)	1.5 (1–2.4)	1.6 (1.1–2.3)	<0.001	<0.001	<0.001	0.493	0.124	0.737	0.001	0.021	0.022	0.071	0.617
Triglyceride (mmol/L)	0.9 (0.3–5.6)	1.1 (0.3–5.6)	0.8 (0.5–2.4)	0.9 (0.4–2.8)	0.8 (0.5–1.8)	0.8 (0.3–1.5)	<0.001	<0.001	0.212	0.149	0.343	0.123	0.010	0.002	0.751	0.572	0.650
VAI	1.21 (0.39–10.89)	1.46 (0.39–10.89)	1.31 (0.43–4.39)	1.00 (0.41–4.79)	1.01 (0.4–3.21)	0.85 (0.32–1.87)	<0.001	<0.001	0.013	0.117	0.244	0.331	0.003	0.001	0.289	0.124	0.803

Results are given as median (minimum–maximum); *p*1: Total PCOS cases vs. Controls, *p*2: Phenotype A vs. Controls, *p*3: Phenotype B vs. Controls, *p*4: Phenotype C vs. Controls, *p*5: Phenotype D vs. Controls, *p*6: Phenotype A vs. Phenotype B, *p*7: Phenotype A vs. Phenotype C, *p*8: Phenotype A vs. Phenotype D, *p*9: Phenotype B vs. Phenotype C, *p*10: Phenotype B vs. Phenotype D, *p*11: Phenotype C vs. Phenotype D. BMI: body mass index, WC: waist circumference, HOMA-IR: Homeostasis Model Assessment of Insulin Resistance, HDL: high-density lipoprotein, VAI: visceral adiposity index.

**Table 2 medicina-61-01673-t002:** The relationship between visceral adiposity index levels and the presence of metabolic syndrome.

	Total PCOS *n* = 180	Phenotype A, *n* = 96	Phenotype B, *n* = 19	Phenotype C, *n* = 35	Phenotype D *n* = 30	Controls *n* = 51	*p*1	*p*2	*p*3	*p*4	*p*5
Presence of MetS, *n* (%)	38 (21.1)	30 (31.3) *^§^	3 (15.8)	2 (5.7) *	3 (10) ^§^	0 (0)	<0.001	<0.001	0.004	0.084	0.021

*p*1: Total PCOS cases vs. Controls, *p*2: Phenotype A vs. Controls, *p*3: Phenotype B vs. Controls, *p*4: Phenotype C vs. Controls, *p*5: Phenotype D vs. Controls. For comparisons between other groups: Phenotype A vs. B *p* = 0.173, * Phenotype A vs. C *p* = 0.003, ^§^ Phenotype A vs. D *p* = 0.021, Phenotype B vs. C *p* = 0.222, Phenotype B vs. D *p* = 0.547, Phenotype C vs. D *p* = 0.517. Mets: metabolic syndrome.

**Table 3 medicina-61-01673-t003:** The relationship between visceral adiposity index levels and some parameters.

	**Spearman’s Rho**	** *p* **
Age	0.214	0.001
Glucose	0.077	0.246
HOMA-IR	0.348	<0.001

HOMA-IR: Homeostasis Model Assessment of Insulin Resistance.

## Data Availability

The data presented in this study are available on request from the corresponding author. The data are not publicly available due to privacy or ethical restrictions.

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
