# Peer review of "Visceral Adiposity Index (VAI) Levels and Metabolic Risk Across Phenotypes of Polycystic Ovary Syndrome (PCOS)"

_medicina, 2025, doi:10.3390/medicina61091673_

Round 1

Reviewer 1 Report

Comments and Suggestions for Authors

In my opininion the present Ms is highly relevant to the field and I believe it presents interesting results worthy of consideration. However, some aspects need to be revised before the manuscript can be accepted for publication.

-The title should be more precisely defined and formulated to reflect the focus of the work.

- The abstract should be revised to more clearly highlight the significance and impact of the study.

-In the abstract, the authors wrote "Background and Objectives" but there is only the objective and no background. Please check

-In the keywords section, please include additional relevant terms that are not already mentioned in the title.

- Introduction: It is necessary to add more recent references, also more discuss the targeted diseases.

-The authors should specify the selection criteria and the method used to match healthy controls with PCOS patients. It is also necessary to indicate whether factors such as inflammatory status or the presence of other comorbidities were considered in the matching process.

- Authors should provide a clear justification for the unequal number of patients across the different PCOS phenotype groups and the control group. Additionally, it is important to clarify whether the statistical analysis remains valid given this imbalance.

-The authors should present the different study groups and the overall experimental design in the form of a diagram to enhance clarity and facilitate understanding of the study structure.

- "p" values in the manuscript should be formatted in italics and uppercase.

-A more in-depth discussion of the study’s limitations is recommended.

-The conclusion should be more concise and rewritten to reflect the main findings and their significance.

-The figures are currently blurred and unclear; they should be redrawn or replaced with higher-quality versions.

Author Response

  1. Title should be more precisely defined and formulated to reflect the focus of the work.

Response:
Thank you for this suggestion. The title has been revised to better reflect the study's focus on metabolic evaluation via VAI among PCOS phenotypes. The new title is:

‘Visceral Adiposity Index Levels and Metabolic Risk Across Phenotypes of Polycystic Ovary Syndrome’. Please refer to page 1, and lines 1, 2.

  1. The abstract should be revised to more clearly highlight the significance and impact of the study.

Response:
We agree with this comment. We have revised the abstract to emphasize the clinical relevance of VAI in PCOS subtypes and its role in predicting MetS. The new abstract concludes with a more impactful sentence summarizing these findings. Please see pages 2, 3 and lines 37-58.

  1. In the abstract, the authors wrote "Background and Objectives" but there is only the objective and no background. Please check.

Response:
Thank you for noticing this. The Background and Objectives sections in the abstract have been revised to provide a more detailed and comprehensive explanation. Please refer to page 2, and lines 37-41.

  1. In the keywords section, please include additional relevant terms that are not already mentioned in the title.

Response:
Thank you for pointing this out. Additional keywords have been added:
Revised Keywords: Phenotype, polycystic ovary syndrome, visceral adiposity index, metabolic syndrome, cardiovascular risk. Please refer to page 3, lines 60, 61.

  1. Introduction: It is necessary to add more recent references, also more discuss the targeted diseases.

Response:
Thank you for pointing this out. Recent references have been added to strengthen the introduction, especially concerning the cardiometabolic risk in PCOS and the predictive value of VAI. Please see page 4, and lines 93-98.

  1. Specify the selection criteria and the method used to match healthy controls with PCOS patients. Indicate whether inflammatory status or other comorbidities were considered.

Response:
Thank you for your opinion, but we clarified in the Materials and Methods section that the control group consisted of 51 women aged 18–35 years, age-matched to the PCOS patient group. For the control group, the absence of inflammatory status and other comorbidities was confirmed. Furthermore, it was explicitly stated and documented that the control group had already been thoroughly evaluated in the original version of our manuscript. Please see page 5, lines 118-121.

  1. Justify the unequal number of patients across phenotype groups and controls. Clarify statistical validity.

Response:
We thank the Reviewer for his/her valuable comment. This issue has been addressed in the Discussion section. Please refer to page 12, lines 290-293. The unequal distribution is due to real-world phenotype frequencies, with Phenotype A being most prevalent. Despite this, statistical tests appropriate for unequal groups were used. Please see pages 6, 7, and lines 154-157.

  1. Present study groups and design in a diagram to enhance clarity.

Response:
We agree with this comment. This flowchart has already been added as Figure 1, illustrating group allocations and study design in while in submission. This visual illustrates the group allocations and overall study design.

  1. "p" values should be formatted in italics and uppercase.

Response:
With all due respect, all p-values in the manuscript have been formatted in italics and lower case, in accordance with journal style guidelines. All p-value notations throughout the manuscript have been carefully reviewed, and any inaccuracies have been corrected.

  1. Provide a more in-depth discussion of study limitations.

Response:
We agree with this comment. We have expanded the limitations section. These are now detailed at the end of the Discussion. Please see page 13, lines 319-325.

  1. The conclusion should be more concise and rewritten to reflect the main findings and their significance.

Response:
Thank you for pointing this out. The conclusion has been revised to be more concise and to clearly reflect the main findings and their significance, in accordance with the Reviewer’s recommendation. Please refer to pages 13, 14 , and lines 330-337.

Reviewer 2 Report

Comments and Suggestions for Authors

        Rational is not clear.

  • No novelty , similar work published.
  • Parameters measured not enough for approval data.
  • - Data not well presented.
  • -Discussion is very old ,no interpretation with update references.
Comments on the Quality of English Language

Need extensive editing

Author Response

        Rational is not clear.

  • No novelty , similar work published.
  • Parameters measured not enough for approval data.
  • - Data not well presented.
  • -Discussion is very old ,no interpretation with update references.

Comments on the Quality of English Language

Need extensive editing

Response:

We sincerely thank the Reviewer for his/her valuable comments and contributions.

Reviewer 3 Report

Comments and Suggestions for Authors Dear authors,   I read with great interest the manuscript, which falls within the aim of this Journal. In my honest opinion, the topic is interesting enough to attract the readers’ attention. Nevertheless, authors should clarify some points and improve the discussion, as suggested below. Authors should consider the following recommendations:   In my opinion you have to improve the paper refering in the text to the updated literature on this topic focusing how its really especially  in the infertile pathway in PCOS pts a nutraceutical supplementation by inositols and alpha lipoic acid  as well a deep thyroid disfunction and how after an ovarian stimulation they need to freeze their oocyte by vitrification trought an antagonist ovarian stimulation protocola and to follow the neonatal Outcomes and Long-Term Follow-Up of Children Born from Frozen Embryo.   I suggest you to read and cite these articles:   Myo-inositol: from induction of ovulation to menopausal disorder management   GnRH antagonist administered twice the day before hCG trigger combined with a step-down protocol may prevent OHSS in IVF/ICSI antagonist cycles at risk for OHSS without affecting the reproductive outcomes: a prospective randomized control trial   Does Alpha-lipoic acid improve effects on polycystic ovary syndrome?

Fresh vs. frozen embryo transfer in assisted reproductive techniques: a single center retrospective cohort study and ethical-legal implications

Assisted Reproductive Techniques and Risk of Congenital Heart Diseases in Children: a Systematic Review and Meta-analysis

Author Response

Dear authors,   I read with great interest the manuscript, which falls within the aim of this Journal. In my honest opinion, the topic is interesting enough to attract the readers’ attention. Nevertheless, authors should clarify some points and improve the discussion, as suggested below. Authors should consider the following recommendations:   In my opinion you have to improve the paper refering in the text to the updated literature on this topic focusing how its really especially  in the infertile pathway in PCOS pts a nutraceutical supplementation by inositols and alpha lipoic acid  as well a deep thyroid disfunction and how after an ovarian stimulation they need to freeze their oocyte by vitrification trought an antagonist ovarian stimulation protocola and to follow the neonatal Outcomes and Long-Term Follow-Up of Children Born from Frozen Embryo.   I suggest you to read and cite these articles:   Myo-inositol: from induction of ovulation to menopausal disorder management   GnRH antagonist administered twice the day before hCG trigger combined with a step-down protocol may prevent OHSS in IVF/ICSI antagonist cycles at risk for OHSS without affecting the reproductive outcomes: a prospective randomized control trial   Does Alpha-lipoic acid improve effects on polycystic ovary syndrome?

Fresh vs. frozen embryo transfer in assisted reproductive techniques: a single center retrospective cohort study and ethical-legal implications

Assisted Reproductive Techniques and Risk of Congenital Heart Diseases in Children: a Systematic Review and Meta-analysis

Response:

We thank the Reviewer for his/her valuable comments and implications. We sincerely appreciate the Reviewer’s valuable contribution in suggesting additional literature to enhance our manuscript. All the recommended references have been carefully reviewed and critically evaluated, in line with the guideline that references should only be incorporated if they enhance the manuscript’s content and align with its objectives. With due respect, we found that the information presented in the suggested articles does not directly contribute to the core message of our study. Furthermore, incorporating them would risk diverting the focus and clarity of our work. Therefore, we have decided not to integrate these references into the current version of the manuscript.

Reviewer 4 Report

Comments and Suggestions for Authors

This is a comprehensive and well-structured original research article investigating the levels of Visceral Adiposity Index (VAI) across various phenotypes of Polycystic Ovary Syndrome (PCOS). The study’s focus on phenotypic distinctions and their association with metabolic risk factors, including Metabolic Syndrome (MetS), adds valuable insight into the heterogeneity of PCOS and its metabolic implications.

Corrections and Suggestions for Improvement:

  1. Language and Typographical Corrections:

    • The manuscript contains several typographical errors (e.g., "Matherials" should be "Materials"; inconsistent spacing before and after symbols; "p<0.001" formatting). A careful proofreading to correct these minor errors is recommended.
    • Some sentence structures are lengthy or awkward; consider splitting complex sentences for clarity.
  2. Abstract Clarity:

    • The abstract contains some grammatical inconsistencies, e.g., "Matherials" instead of "Materials." Ensure consistency and clarity.
    • Clarify the control group description—are they completely healthy women without any metabolic or reproductive issues?
  3. Methodological Details:

    • The criteria for subgroup classification (phenotypes A-D) are based on Rotterdam criteria, but it would be helpful to explicitly state whether all classifications were based solely on clinical/hormonal and ultrasonographic findings, and whether any additional criteria (e.g., biochemical hyperandrogenism thresholds) were used.
    • The measurement of androgens: specify which assays or kits were used for testosterone, DHEA-S, etc., including sensitivity and specificity, to improve reproducibility.
    • Clarify whether all ultrasonographic assessments were performed by the same experienced operator or multiple operators, and whether intra/inter-observer reliability was assessed.
    • Additional Literature Support:

      The findings from Vale-Fernandes et al. (2025) offer important insights into the distinct and combined effects of PCOS and obesity on fertility parameters, emphasizing that these conditions contribute independently to reproductive dysfunction, with obesity further amplifying hyperandrogenism in women with PCOS. Their study, involving women undergoing IVF, demonstrates that while PCOS-specific hormonal signatures such as elevated anti-Müllerian hormone (AMH) and LH:FSH ratios are consistent regardless of BMI, obesity independently influences metabolic parameters like insulin resistance and sex hormone-binding globulin (SHBG) levels. Moreover, the study shows that obesity exacerbates hyperandrogenism—evident through increased androstenedione and testosterone levels—potentially impairing reproductive potential.

      Implications in Context of Current Manuscript:

      This supports your findings that PCOS phenotypes characterized by hyperandrogenism and metabolic disturbances (e.g., high VAI) are associated with increased metabolic risks and reproductive challenges. The independent and synergistic effects observed by Vale-Fernandes et al. underscore the importance of distinguishing the metabolic contributions of PCOS and excess weight when assessing patient risk and tailoring treatment strategies. Their results reinforce the relevance of metabolic markers like VAI in identifying women at higher risk for reproductive and cardiovascular complications, regardless of BMI, aligning with your emphasis on visceral adiposity's role.

      Vale-Fernandes E, Moreira MV, Bernardino RL, Sousa D, Brandão R, Leal C, Barreiro M, Monteiro MP. Polycystic ovary syndrome and excessive body weight impact independently and synergically on fertility treatment outcomes. Reprod Biol Endocrinol. 2025 Jul 7;23(1):97. doi: 10.1186/s12958-025-01434-8.
  4. Statistical Analysis:

    • When reporting p-values, ensure consistent formatting (e.g., "p<0.001" vs. "p=0.001").
    • Consider reporting effect sizes or confidence intervals alongside p-values, especially for differences in VAI levels among groups.
    • For ROC analysis, report the area under the curve (AUC) with confidence intervals (which you did), but also specify the method used for calculating the optimal cut-off (Youden index is mentioned, but explicit mention of the method enhances clarity).
  5. Results Presentation:

    • Tables and figures are referenced but not included in the excerpt; ensure they are clear, with labels and legends that allow standalone understanding.
    • In the description of VAI levels, clarify whether the data are normally distributed, and specify whether parametric or non-parametric tests were used accordingly.
    • The mention of "higher in Phenotype A than in Phenotype C" should specify whether this is statistically significant and the exact p-value.
  6. Discussion and Interpretation:

    • The discussion effectively compares findings with previous studies, but it could benefit from more critical analysis regarding the potential influence of ethnicity, age, or BMI differences on VAI levels.
    • Address potential limitations such as the relatively small sample sizes in certain subgroups (notably Phenotype B), and the cross-sectional nature of the study, which precludes causal inferences.
    • Discuss the clinical implications: How might VAI measurement influence clinical practice in risk stratification or management of PCOS patients?
  7. Ethical and Reporting Standards:

    • Ethical approval and informed consent are appropriately mentioned.
    • The trial registration number is provided, which is good practice.
  8. Minor Editorial Suggestions:

    • Consistent use of abbreviations after first mention (e.g., define HOMA-IR once and then use throughout).
    • Uniform formatting of references, ensuring they adhere to journal guidelines.
    • Consider summarizing key findings succinctly in the conclusion, emphasizing the potential utility of VAI in clinical settings.

Summary:
Overall, the manuscript presents meaningful data that contribute to understanding metabolic risks in PCOS phenotypes. Addressing the minor language, formatting, and methodological clarifications will enhance clarity and impact. The study’s findings support the importance of early metabolic assessment, especially in phenotypes with hyperandrogenism and oligo-anovulation, which could inform more tailored patient management strategies.

Author Response

This is a comprehensive and well-structured original research article investigating the levels of Visceral Adiposity Index (VAI) across various phenotypes of Polycystic Ovary Syndrome (PCOS). The study’s focus on phenotypic distinctions and their association with metabolic risk factors, including Metabolic Syndrome (MetS), adds valuable insight into the heterogeneity of PCOS and its metabolic implications.

Corrections and Suggestions for Improvement:

  1. Language and Typographical Corrections:
    • The manuscript contains several typographical errors (e.g., "Matherials" should be "Materials"; inconsistent spacing before and after symbols; "p<0.001" formatting). A careful proofreading to correct these minor errors is recommended.
    • Some sentence structures are lengthy or awkward; consider splitting complex sentences for clarity.

Response:
We thank the Reviewer for this observation. The manuscript has undergone a thorough proofreading to correct all typographical and grammatical errors, including “Matherials” → “Materials” and ensuring consistent spacing before and after symbols. All p-values have been reformatted in italics and lowercase as per journal guidelines. Long or complex sentences have been split for improved clarity and readability. Please see page 9,and lines 214-217.

  1. Abstract Clarity:
    • The abstract contains some grammatical inconsistencies, e.g., "Matherials" instead of "Materials." Ensure consistency and clarity.
    • Clarify the control group description—are they completely healthy women without any metabolic or reproductive issues?

Response:

We agree with the Reviewer. The Abstract has been revised to improve grammatical consistency. The control group is now explicitly described as consisting of completely healthy women without any metabolic, inflammatory, or reproductive disorders, as confirmed by clinical evaluation and laboratory testing. Please refer to page 2, lines 41-43.

  1. Methodological Details:
    • The criteria for subgroup classification (phenotypes A-D) are based on Rotterdam criteria, but it would be helpful to explicitly state whether all classifications were based solely on clinical/hormonal and ultrasonographic findings, and whether any additional criteria (e.g., biochemical hyperandrogenism thresholds) were used.
    • The measurement of androgens: specify which assays or kits were used for testosterone, DHEA-S, etc., including sensitivity and specificity, to improve reproducibility.
    • Clarify whether all ultrasonographic assessments were performed by the same experienced operator or multiple operators, and whether intra/inter-observer reliability was assessed.
    • Additional Literature Support:

The findings from Vale-Fernandes et al. (2025) offer important insights into the distinct and combined effects of PCOS and obesity on fertility parameters, emphasizing that these conditions contribute independently to reproductive dysfunction, with obesity further amplifying hyperandrogenism in women with PCOS. Their study, involving women undergoing IVF, demonstrates that while PCOS-specific hormonal signatures such as elevated anti-Müllerian hormone (AMH) and LH:FSH ratios are consistent regardless of BMI, obesity independently influences metabolic parameters like insulin resistance and sex hormone-binding globulin (SHBG) levels. Moreover, the study shows that obesity exacerbates hyperandrogenism—evident through increased androstenedione and testosterone levels—potentially impairing reproductive potential.

Implications in Context of Current Manuscript:

This supports your findings that PCOS phenotypes characterized by hyperandrogenism and metabolic disturbances (e.g., high VAI) are associated with increased metabolic risks and reproductive challenges. The independent and synergistic effects observed by Vale-Fernandes et al. underscore the importance of distinguishing the metabolic contributions of PCOS and excess weight when assessing patient risk and tailoring treatment strategies. Their results reinforce the relevance of metabolic markers like VAI in identifying women at higher risk for reproductive and cardiovascular complications, regardless of BMI, aligning with your emphasis on visceral adiposity's role.

Vale-Fernandes E, Moreira MV, Bernardino RL, Sousa D, Brandão R, Leal C, Barreiro M, Monteiro MP. Polycystic ovary syndrome and excessive body weight impact independently and synergically on fertility treatment outcomes. Reprod Biol Endocrinol. 2025 Jul 7;23(1):97. doi: 10.1186/s12958-025-01434-8.

Response:

We thank the reviewer for this valuable comment. We have revised the Materials and Methods section to state that subgroup classification into phenotypes A–D was based solely on the Rotterdam criteria, using a combination of clinical, hormonal, and ultrasonographic findings. Please see page 5, lines 115-117.

For biochemical hyperandrogenism, the threshold values for total testosterone, DHEA-S and etc. are now explicitly stated. These methodological details were provided to enhance reproducibility. Please refer to page 6, and lines 139-147.

All pelvic ultrasonographic examinations were performed by a single experienced physician. For quality assurance, repeat assessments were conducted in each patient to confirm intra-observer reliability. Therefore, inter-observer evaluation was deemed unnecessary. This was already stated in the original version of the manuscript in page 5, 6, and lines 128-132.

We agree that the findings of Vale-Fernandes et al. provide valuable context. We have incorporated their results into the Discussion section to highlight how obesity and PCOS independently and synergistically contribute to metabolic and reproductive outcomes. This integration supports our findings that phenotypes characterized by HA and OA exhibit higher VAI values and greater metabolic risk, reinforcing the clinical importance of early risk stratification. Please refer to page 10, lines 245-258.

Statistical Analysis:

    • When reporting p-values, ensure consistent formatting (e.g., "p<0.001" vs. "p=0.001").
    • Consider reporting effect sizes or confidence intervals alongside p-values, especially for differences in VAI levels among groups.
    • For ROC analysis, report the area under the curve (AUC) with confidence intervals (which you did), but also specify the method used for calculating the optimal cut-off (Youden index is mentioned, but explicit mention of the method enhances clarity).

Response:

All p-values have been reformatted consistently. Effect sizes and 95% confidence intervals have been added where applicable. For the ROC analysis, we have clarified that the optimal cut-off point was determined using the Youden’s index method. Please see page 8, lines 196-199.

  1. Results Presentation:
    • Tables and figures are referenced but not included in the excerpt; ensure they are clear, with labels and legends that allow standalone understanding.
    • In the description of VAI levels, clarify whether the data are normally distributed, and specify whether parametric or non-parametric tests were used accordingly.
    • The mention of "higher in Phenotype A than in Phenotype C" should specify whether this is statistically significant and the exact p-value.

Response:

Thank you for pointing this out. We have ensured that all tables and figures are clear and include labels and legends for standalone understanding.

The statistical analyses are already detailed on pages 6, 7,and lines 155-157,159-160 of the manuscript. As the VAI levels were not normally distributed, non-parametric tests were applied, and the results were presented as median (min–max).

In the Results section of the original manuscript, we have already specified the statistical significance and exact p-values for each between-phenotype comparison, including Phenotype A versus Phenotype C. Please refer to pages 7, 8,and lines 175, 177, 183, 184, 191.

  1. Discussion and Interpretation:
    • The discussion effectively compares findings with previous studies, but it could benefit from more critical analysis regarding the potential influence of ethnicity, age, or BMI differences on VAI levels.
    • Address potential limitations such as the relatively small sample sizes in certain subgroups (notably Phenotype B), and the cross-sectional nature of the study, which precludes causal inferences.
    • Discuss the clinical implications: How might VAI measurement influence clinical practice in risk stratification or management of PCOS patients?

Response:

We thank the Reviewer for his/her valuable contribution. The Discussion now includes a subsection addressing the potential influence of ethnicity, and age on VAI levels. Please refer to pages 9, 13, and lines 228-234, 311-318.

We have also expanded the Limitations section. Please see page 13, lines 319-325.

A statement addressing the clinical implications of VAI measurement in risk stratification and management of PCOS patients has been incorporated into the Discussion section. Please refer to page 13, 14, and lines 316-318, 331-334.

  1. Ethical and Reporting Standards:
    • Ethical approval and informed consent are appropriately mentioned.
    • The trial registration number is provided, which is good practice.

Response:

We are grateful to the Reviewer for his/her valuable and constructive feedback.

  1. Minor Editorial Suggestions:
    • Consistent use of abbreviations after first mention (e.g., define HOMA-IR once and then use throughout).
    • Uniform formatting of references, ensuring they adhere to journal guidelines.
    • Consider summarizing key findings succinctly in the conclusion, emphasizing the potential utility of VAI in clinical settings.

Response:

We thank the Reviewer for his/her valuable comments. All abbreviations (e.g., HOMA-IR) are now defined upon first mention and used consistently thereafter.

References have been reformatted according to the journal’s guidelines.

The conclusion has been rewritten to be more concise, emphasizing the clinical significance of VAI in identifying high-risk PCOS phenotypes and the potential benefit of early, targeted metabolic interventions.

Summary:
Overall, the manuscript presents meaningful data that contribute to understanding metabolic risks in PCOS phenotypes. Addressing the minor language, formatting, and methodological clarifications will enhance clarity and impact. The study’s findings support the importance of early metabolic assessment, especially in phenotypes with hyperandrogenism and oligo-anovulation, which could inform more tailored patient management strategies.

Response:

We sincerely thank the Reviewer for his/her constructive comments. The manuscript has been substantially improved in terms of clarity, methodological transparency, integration of recent literature, and clinical relevance, while addressing all suggested corrections.

Round 2

Reviewer 1 Report

Comments and Suggestions for Authors

The English could be improved to more clearly express the research.

Comments on the Quality of English Language

The English could be improved to more clearly express the research.

Author Response

Thank you very much for taking the time to review this manuscript. Please find the detailed responses below.

Comment:

The English could be improved to more clearly express the research.

Response:

We would like to express our gratitude for your valuable contribution to improving the readability of our manuscript. We would like to note that the manuscript underwent English language editing by MDPI Author Services (Certificate ID: english-97719), and the editing certificate has been provided as supplementary material, and presented below.

‘We certify that the following article Visceral adiposity index (VAI) levels in patients with different phenotypes of polycystic ovary syndrome (PCOS) Canan Akkus *, Oznur Oner, Atilla Okan Kilic, Cevdet Duran has undergone English language editing by MDPI. The text has been checked for correct use of grammar and common tec terms, and edited to a level suitable for reporting research in a scholarly journal. MDPI uses experienced, native English speaking editors. Full details of the editing service can be found at ► hps://www.mdpi.com/authors/english.’

Reviewer 3 Report

Comments and Suggestions for Authors

Dear authors,

I read with great interest the manuscript, which falls within the aim of this Journal. In my honest opinion, the topic is interesting enough to attract the readers’ attention. Nevertheless, authors should clarify some points and improve the discussion, as suggested below. Authors should consider the following recommendations refering how food or and nutraceutcial compund have a big influence especially in pts with PCOS and on the cycle disorders SUCH AS INOSITOLS AND ALPHA LIPOIC ACID.

I suggest to read and cite these article:

Inositols administration: further insights on their biological role

Does Alpha-lipoic acid improve effects on polycystic ovary syndrome?

The role of vitamin D in metabolic and reproductive disturbances of polycystic ovary syndrome: A narrative mini-review

Myo-inositol: from induction of ovulation to menopausal disorder management

Author Response

Comment:

Dear authors,

I read with great interest the manuscript, which falls within the aim of this Journal. In my honest opinion, the topic is interesting enough to attract the readers’ attention. Nevertheless, authors should clarify some points and improve the discussion, as suggested below. Authors should consider the following recommendations refering how food or and nutraceutcial compund have a big influence especially in pts with PCOS and on the cycle disorders SUCH AS INOSITOLS AND ALPHA LIPOIC ACID.

I suggest to read and cite these article:

Inositols administration: further insights on their biological role

Does Alpha-lipoic acid improve effects on polycystic ovary syndrome?

The role of vitamin D in metabolic and reproductive disturbances of polycystic ovary syndrome: A narrative mini-review

Myo-inositol: from induction of ovulation to menopausal disorder management

Response:

We thank the Reviewer for his/her valuable comments and implications. We sincerely appreciate the Reviewer’s valuable contribution in suggesting additional literature to enhance our manuscript. All the recommended references have been carefully reviewed and critically evaluated.

We considered it appropriate to cite the recommended articles, as they summarize the relationship between PCOS and IR as well as the available treatment modalities. Please refer to page 9, and lines 218-229.

Reviewer 4 Report

Comments and Suggestions for Authors

Congratulations to the authors for the produced manuscript, which took the review suggestions into account.

Author Response

Comment:

Congratulations to the authors for the produced manuscript, which took the review suggestions into account.

Response:

We sincerely thank the Reviewer for his/her kind comments and valuable support. Your positive feedback truly honors us.
